# From Uncertainty to Consent: Educational Intervention Effects on Knowledge and Willingness to Donate Organs After Death

**DOI:** 10.3390/healthcare13192483

**Published:** 2025-09-30

**Authors:** Aruzhan Asanova, Saule Shaisultanova, Dana Anafina, Gulnur Daniyarova, Vitaliy Sazonov, Aidos Bolatov, Aigerim Abdiorazova, Yuriy Pya

**Affiliations:** 1Department of Science, “University Medical Center” Corporate Fund, Astana 010000, Kazakhstan; asanova.aruzhan@umc.org.kz (A.A.); anafinad@msu.edu (D.A.); daniyarova.g@umc.org.kz (G.D.); 2Department of Medical and Regulatory Affairs, “University Medical Center” Corporate Fund, Astana 010000, Kazakhstan; shaisultanova.s@umc.org.kz; 3Pediatric Intensive Care Unit, “University Medical Center” Corporate Fund, Astana 010000, Kazakhstan; dr.sazonov@gmail.com; 4Department of Surgery, School of Medicine, Nazarbayev University, Astana 010000, Kazakhstan; 5Shenzhen University Medical School, Shenzhen University, Shenzhen 518060, China; 6Division of Strategic Development and Science, “Human Research & Development” LLP, Astana 010000, Kazakhstan; 7Management Board, “University Medical Center” Corporate Fund, Astana 010000, Kazakhstan; a.aigerim@umc.org.kz (A.A.); yuriy.pya@umc.org.kz (Y.P.)

**Keywords:** organ donation, postmortem consent, educational intervention, knowledge, attitudes, barriers

## Abstract

**Background:** The willingness to donate organs after death remains low in many populations, often due to informational and psychological barriers. This study assessed the impact of an educational lecture on knowledge and attitudes toward postmortem organ donation among university students in Kazakhstan. **Methods:** A total of 129 students completed a pre-lecture questionnaire on donation attitudes, knowledge, and barriers; 97 also completed the post-lecture assessment. Changes were analyzed using paired *t*-tests, repeated-measures ANOVA, and logistic regression. Participants were grouped by attitudinal changes to identify predictors of consent. **Results:** Knowledge about organ donation increased significantly after the lecture (*p* < 0.001), with larger gains among females and non-medical students. The number of participants who were willing to donate rose from 27 to 56 (*p* < 0.001). About 37% showed a positive shift, while 3% shifted toward refusal. In the initially ambivalent group (*n* = 49), female gender (AOR = 35.6), greater knowledge gain (AOR = 3.03), and lower perceived barriers (AOR = 0.05) predicted a change towards consent. Uncertainty about how to express consent was the only significantly differing barrier (*p* = 0.036). **Conclusion:** A brief educational lecture effectively increased knowledge and willingness to donate. Targeted information on procedural aspects may reduce indecision and promote informed donor registration.

## 1. Introduction

Despite global advances in transplantation medicine, the persistent gap between organ supply and demand remains a pressing public health issue [1]. In many countries, including Kazakhstan, low rates of donor registration and family consent significantly limit the availability of organs for transplantation [2]. Internationally, the absence of standardized criteria for brain death and organ donation has also been identified as a major barrier, contributing not only to the loss of potential donors but also to the rise in organ trafficking as a multibillion dollar global industry [3]. One of the key barriers to organ donation is a lack of public awareness, coupled with psychological barriers such as fear, religious misconceptions, and procedural uncertainty [4,5,6]. These factors are particularly pronounced among young people, who, despite being more receptive to social causes, often lack formal education or personal experience with the donation process.

Previous studies have consistently shown that knowledge is a critical predictor of willingness to donate organs after death [7,8]. Similar associations have been demonstrated in other domains of health behavior, such as vaccine uptake, where literacy and targeted education were found to significantly shape attitudes among Kazakhstani students [9]. Educational interventions—especially those targeted at younger cohorts—have shown potential in correcting misconceptions and fostering more favorable attitudes toward donation [10]. Recent Spanish studies have further demonstrated the effectiveness of educational interventions across different age groups. For example, randomized trials and school-based programs among university students and adolescents showed that structured training, creative approaches such as short film projects, nurse-led workshops, and social media significantly improved knowledge, corrected misconceptions, and fostered more favorable attitudes toward donation [11,12,13]. However, evidence from countries with low donation rates, such as Japan, indicates that while greater knowledge may be linked to increased willingness to donate, it does not necessarily translate into actual donor registration [14]. The effectiveness of such interventions is context dependent, and few studies have evaluated their impact in Central Asian populations, where cultural, linguistic, and religious factors may uniquely influence decision-making [15]. Thus, broader surveys among university students in Kazakhstan similarly highlight how knowledge, cultural beliefs, and contextual factors affect their engagement in health-promoting behaviors [16].

Kazakhstan provides a particularly compelling setting for such research. Although a legal framework for organ donation exists, public skepticism and limited awareness of the registration process have hindered donor enrollment, while moral and religious perceptions further complicate outreach [4,5,17]. The country follows an opt-in system, requiring individuals to actively register their decision through healthcare facilities or the national e-government portal. If no decision is recorded, families decide on behalf of the deceased, making communication of one’s wishes critical. As of 2024, only 111.6 thousand citizens had documented their preferences, 104.4 thousand opting out and just 7.2 thousand consenting, underscoring the urgent need to strengthen both registration and family discussions, as family objections remain a major barrier in Kazakhstan and globally [4].

The purpose of this study was to evaluate the effect of a single, standardized educational presentation on knowledge and attitudes toward postmortem organ donation among university students at three major institutions in Kazakhstan. In addition to assessing pre- and post-intervention changes in knowledge and donation decisions, the study examined key demographic and psychosocial predictors of attitude change, with a focus on the ambivalent “decision left to relatives” group. By identifying modifiable barriers and responsive subgroups, the findings provide practical insights for designing scalable, culturally sensitive interventions that promote informed decision-making about organ donation.

## 2. Materials and Methods

### 2.1. Study Design and Participants

This study employed a quasi-experimental pre-test–post-test design to evaluate the impact of a single educational lecture on university students’ knowledge, attitudes, and decision-making regarding postmortem organ donation (Figure 1A). The questionnaire was available in both Kazakh and Russian to ensure accessibility across language groups. The study was conducted at three universities in Kazakhstan: Nazarbayev University (NU), L.N. Gumilyov Eurasian National University (ENU), and Astana Medical University (AMU). Convenience sampling was used to recruit undergraduate students from both medical and non-medical programs. All medical students were enrolled in the General Medicine/MBBS or MD programs.

A total of 129 students completed the pre-test questionnaire, and 97 of them also completed the post-test questionnaire. Thus, while all post-test participants had completed the pre-test, not all pre-test participants were retained for follow-up. Participation was voluntary, and informed consent was obtained prior to data collection.

### 2.2. Rationale for Student Sample Selection

University students were selected as the target population for this study due to their transitional developmental stage, which is often characterized by increased openness to new information, identity formation, and civic engagement. This demographic is also more likely to be exposed to public health messaging in digital and institutional environments and is thus a key group for shaping future societal attitudes toward organ donation. Moreover, students represent a highly accessible and diverse population in terms of educational background, religious affiliation, and socioeconomic status, allowing for nuanced subgroup analyses. Importantly, although young adults are not the primary candidates for immediate donor registration, their early exposure to accurate information can have long-term implications for donor behavior, family discussions, and advocacy. By targeting this group, the intervention aimed not only to assess the immediate impact of educational content but also to inform scalable strategies for integrating donation-related education into academic settings.

### 2.3. Procedure

Participants first completed a structured self-administered questionnaire assessing sociodemographic characteristics, baseline knowledge of organ donation, perceived barriers, and their current donation decision (consent, refusal, or decision left to relatives). They then attended a standardized 60-min lecture focused on medical, ethical, legal, and procedural aspects of postmortem organ donation in Kazakhstan (Appendix A). All survey instruments underwent content validation by a multidisciplinary panel of experts in transplant coordination, transplantation surgery, public health, medical education, and social psychology. This process ensured cultural and contextual appropriateness before data collection.

The content of the educational intervention was standardized across all three universities. Lectures were delivered in Kazakh and Russian, and organized into two segments: a structured informational presentation followed by an interactive Q&A session. The intervention was designed in accordance with international recommendations on organ donation and transplantation education [18], which emphasize short, structured, and culturally sensitive activities that involve both healthcare professionals and transplant recipients.

Each session featured three speakers. First, a transplant coordinating expert discussed the current state of organ donation in Kazakhstan, addressed common misconceptions, and clarified consent procedures. Second, a transplantation surgeon provided an overview of the surgical complexities involved and highlighted recent advances in transplantation practices within Kazakhstan. These components reflected broader international guidance that educational activities should combine medical expertise with clear information on donation processes, legal frameworks, and myths. Finally, an organ recipient shared a personal narrative that emphasized the emotional and life-changing aspects of donation. This integration of factual knowledge and lived experience was intended to foster empathy, address cultural and religious concerns, and encourage informed decision-making.

Each lecture concluded with a clear call to action, encouraging participants to engage in conversations about organ donation with their families and close contacts. Throughout the lecture, students were encouraged to ask questions, reflect on their own attitudes, and clarify misconceptions, further reinforcing the interactive nature of the intervention. Immediately after the lecture, participants were asked to complete the questionnaire again, with only knowledge and attitudes reassessed at this stage, allowing for paired pre- and post-intervention comparisons. Each response was linked using either the participant’s email address or a unique self-generated identifier to ensure accurate matching while maintaining confidentiality.

### 2.4. Measures

#### 2.4.1. Sociodemographic Variables

Data were collected on participants’ age, gender, university, academic specialization (medical vs. non-medical), language of instruction (Kazakh or Russian), residence prior to university (urban or rural), religious affiliation, self-rated religiosity (1 = not religious to 5 = very religious), and perceived economic well-being (1 = very low to 5 = very high).

#### 2.4.2. Knowledge

Knowledge of postmortem organ donation was assessed using eight statements with three response options: “True,” “False,” or “Don’t know.” Each item was scored as 1 for a correct response and 0 for an incorrect or “Don’t know” response, resulting in a total knowledge score ranging from 0 to 8, with higher scores indicating greater knowledge. The items were taken from a previously published questionnaire [17], where the full wording of the questions is available.

#### 2.4.3. Barriers

Perceived barriers to donation were measured using a set of Likert-scale items (1 = strongly disagree to 5 = strongly agree) previously published [17]. Items addressed concerns such as religious objections, fear of bodily harm, mistrust in the healthcare system, and lack of information on how to register consent. A mean barrier score was computed for each participant.

#### 2.4.4. Donation Decision

Participants indicated one of three decisions: (1) LC (lifetime consent) to donate their organs; (2) LR (lifetime refusal) to donate; or (3) DLCR (decision left to close relatives). Based on changes between pre- and post-test, participants were later grouped into categories representing their decision trajectories (e.g., DLCR-LC, LC-LC).

### 2.5. Statistical Analysis

Descriptive statistics were computed for all variables. To assess the representativeness of participants retained at follow-up, we compared sociodemographic and psychosocial variables between the pre-test (*n* = 129) and post-test (*n* = 97) groups utilizing independent sample *t*-test. Group differences in donation attitudes at pre-test were examined using χ^2^-tests for categorical variables and one-way ANOVAs for continuous variables, followed by post hoc comparisons where applicable. To evaluate changes in knowledge, paired *t*-tests and repeated measures ANOVA were conducted, including interaction effects with sociodemographic variables. Decision changes were assessed using Bowker’s test of symmetry for paired categorical data. A binary logistic regression was performed on the DLCR subgroup to identify predictors of change toward consent, with independent variables including gender, age, language, university, religiosity, perceived barriers, economic well-being, and change in knowledge. All odds ratios reported from logistic regression analyses represent adjusted odds ratios (AORs), controlling for all covariates included in the model. Statistical significance was set at *p* < 0.05. All analyses were conducted using Jamovi (version 2.6.17) and JASP (version 0.19.3).

### 2.6. Ethical Considerations

This study was conducted in accordance with the principles of the Declaration of Helsinki and received ethical approval from the Local Bioethics Commission of the “University Medical Center” Corporate Fund (Protocol No. 3 dated 14 July 2023) prior to data collection. The study was not preregistered, given its exploratory and educational nature. Following the post-test, participants were provided with a debriefing that included information on the process of organ donor registration in Kazakhstan and resources for further guidance.

Participation was voluntary, and online informed consent was obtained from all participants before the start of the survey. The survey instruments were provided in both Kazakh and Russian languages to respect participants’ language preferences and ensure comprehension.

To ensure confidentiality and accurate data matching, participants were asked to provide either their email address or a unique self-generated identifier known only to them. This approach allowed pre- and post-intervention responses to be linked without compromising anonymity. All data were stored securely and were accessible only to the research team. No personally identifiable information was used in the analysis or reporting of the results.

Participants were informed about the purpose of the study, their right to withdraw at any time without penalty, and the confidentiality of their responses. No financial or academic incentives were provided for participation. The study involved minimal risk, as it focused solely on educational content and self-reported attitudes.

## 3. Results

A total of 129 participants completed the pre-test questionnaire, while 97 of them also completed the post-test questionnaire. Thus, all individuals in the post-test group participated in the pre-test, but not all pre-test respondents were present at the post-test. Descriptive characteristics of the study population at both time points are summarized in Table 1.

As shown in Table 1, participants who completed both the pre- and post-test assessments did not significantly differ from the full baseline sample across sociodemographic, religious, or psychosocial characteristics (all *p* > 0.25), suggesting that attrition was unlikely to have introduced systematic bias. This finding was further supported by Appendix A, which compared baseline characteristics of completers and non-completers. Although some differences were observed by school (*p* < 0.001), specialization, age, and language (*p* < 0.05), overall, the groups were broadly similar across most variables (Appendix A), indicating that selection bias was unlikely to have substantially influenced the main study findings.

At pre-test, the majority of participants were from AMU (45.0%), followed by ENU (33.3%) and NU (21.7%). The gender distribution was skewed toward females (80.6%), with only 19.4% identifying as male. The average age of participants at baseline was 18.8 years (SD = 2.16), with a slight decrease in the post-test group (M = 18.6, SD = 1.69), reflecting minor differences in sample composition.

Regarding academic specialization, 63.6% were medical students and 36.4% were non-medical. Kazakh was the primary language for most participants (77.5%), while it was Russian for 22.5%. In terms of geographical background, 69.8% of participants had resided in urban areas prior to entering university, and 30.2% were from rural areas.

Most participants identified as Muslim (82.2%), while 10.1% identified as atheist and 7.8% as agnostic. The average self-reported religiosity level was 2.90 (SD = 1.12) on a 5-point scale, and perceived economic well-being was 3.57 (SD = 0.86) on a 5-point scale as well. Initial knowledge about organ donation (scored 0 to 8) averaged 5.29 (SD = 1.82), and perceived barriers to organ donation (on a scale of 1–5) averaged 3.14 (SD = 0.66).

### 3.1. Pre-Test Differences in Attitudes Toward Postmortem Organ Donation

Among the 129 participants who completed the pre-test, their stated positions on postmortem organ donation were distributed as follows: 25 participants (19.4%) refused to donate, 64 (49.6%) preferred to leave the decision to close relatives, and 40 (31.0%) expressed lifetime consent. Several significant differences emerged across these three groups based on sociodemographic and psychological factors (Table 2).

Institutional affiliation significantly influenced attitudes (*p* = 0.002), with no participants from NU reporting refusal and the highest proportion of refusal coming from AMU (25.9%). Age also differed significantly across groups (*p* = 0.027), with the consent group being slightly older on average than the refusal or family-decision groups.

Language background was associated with donation attitude (*p* = 0.013), with Russian-speaking participants more likely to express consent (51.7%) compared to Kazakh-speaking participants (25.0%). Although not statistically significant at the conventional threshold, a trend was observed for residence before university (*p* = 0.082), with urban residents more likely to agree to donation (34.4%) than rural participants (23.1%).

Significant differences were also found in knowledge and perceived barriers. Participants who consented to donation had significantly higher initial knowledge scores compared to those who refused or deferred the decision (*p* = 0.033). The most pronounced differences were observed in barriers to organ donation: those who refused had the highest mean barrier score, while those who consented reported significantly fewer perceived barriers (*p* < 0.001).

Although religiosity, economic well-being, and medical specialization did not significantly differ between groups, religiosity approached significance (*p* = 0.094), with higher religiosity associated with greater reluctance toward donation.

Post hoc analyses confirmed that participants who expressed consent had significantly higher knowledge and lower barrier scores than both the refusal group and those who left the decision to family.

### 3.2. Moderators of Knowledge Gain After the Lecture

A paired *t*-test revealed a statistically significant increase in participants’ knowledge about postmortem organ donation following the lecture, with mean scores rising from 5.34 ± 1.67 before to 6.94 ± 1.33 after the intervention (t = 8.74, *p* < 0.001; Figure 1B).

When comparing participants across universities (Table 3), a repeated measures ANOVA showed a significant interaction effect between knowledge level and university (*p* = 0.019). While all groups demonstrated improvement, students from ENU showed the most substantial increase (mean increase from 4.61 ± 1.62 to 6.96 ± 1.17, *p* < 0.001). A post hoc comparison confirmed that students from NU had significantly higher baseline knowledge levels compared to those from ENU and AMU (*p* < 0.001).

Gender also had a moderating effect on knowledge gain, with a significant interaction between gender and knowledge level (*p* = 0.041). While males and females did not differ significantly in baseline knowledge, females demonstrated a larger increase in knowledge post-lecture (*p* < 0.001).

Specialization significantly influenced knowledge levels as well (Figure 1C). Medical students had higher baseline scores than non-medical students and showed smaller knowledge gains overall. The interaction between specialization and knowledge was significant (*p* = 0.007), with non-medical students showing greater improvement (mean change = 2.59 points, *p* < 0.001) compared to medical students (mean change = 1.28, *p* < 0.001).

Language of instruction was also associated with knowledge level. Russian-speaking students had higher baseline and post-lecture scores compared to Kazakh-speaking students, and a significant knowledge gain was observed in both groups (*p* < 0.001 for both). However, the interaction effect was not statistically significant (*p* = 0.290).

No significant differences in knowledge change were found based on place of residence prior to university (urban vs. rural; *p* = 0.541). Similarly, although baseline differences in knowledge by religion were significant (*p* = 0.038), the interaction between religion and knowledge change was not significant (*p* = 0.257).

Finally, knowledge change was not associated with participants’ age (F = 0.0002, *p* = 0.960), level of religiosity (F = 3.13, *p* = 0.08), or subjective economic well-being (F = 2.42, *p* = 0.123).

### 3.3. Decision Trajectories and Attitudinal Shifts Toward Organ Donation

Prior to the lecture, 27 individuals expressed consent to donate their organs after death, 50 chose to leave the decision to their family members, and 20 explicitly refused. After attending the informational lecture, the distribution shifted significantly (Figure 1D). Among those who had previously deferred the decision to family, 23 participants changed to “consent,” and only 1 switched to “refuse.” Similarly, 25 out of 27 participants who initially gave consent maintained their decision, while 2 changed to “refuse.” Overall, the number of participants consenting to organ donation increased from 27 to 56. A generalized McNemar test (Bowker’s test of symmetry) indicated that this change in distribution was statistically significant, χ^2^(3) = 22.7, *p* < 0.001. These results suggest that the lecture had a significant impact on participants’ willingness to donate organs after death.

Based on participants’ pre- and post-lecture decisions regarding postmortem organ donation, they were categorized into seven groups to reflect the direction and nature of their decision change. The coding included “LR” for participants who preferred to leave the decision to relatives, “DLCR” for those who declined donation but left the choice to relatives, and “LC” for those likely to consent to donation.

Among the 97 participants, 5 individuals (5.2%) shifted from leaving the decision to relatives to declining but leaving the choice to relatives (LR-DLCR), while 8 participants (8.2%) moved from leaving the decision to relatives to being likely to consent (LR-LC). A substantial proportion, 23 participants (23.7%), changed from declining but deferring to relatives to expressing a likely willingness to consent (DLCR-LC). Additionally, 26 participants (26.8%) remained in the DLCR category, and 25 (25.8%) consistently expressed willingness to donate (LC-LC). Seven individuals (7.2%) maintained their initial position of leaving the decision to relatives (LR-LR), while only three participants (3.1%) demonstrated a negative shift in attitude, moving from either the consent or DLCR category to leaving the decision to relatives.

Taken together, nearly 37% of participants showed a positive shift toward increased willingness to donate, approximately 35% maintained their initial stance, and only a small fraction (3%) exhibited a decline in donation readiness. These findings suggest that the informational lecture was effective in positively influencing participants’ attitudes toward postmortem organ donation. Moreover, a particularly significant increase in knowledge was observed in the DLCR-LC group (t = 7.24, *p* < 0.001).

To further explore the determinants of positive shifts in donation attitudes, a binary logistic regression was conducted within the DLCR group (*n* = 49), comparing participants who changed their decision to consent (DLCR-LC, *n* = 23) with those who maintained their position (DLCR-DLCR, *n* = 26). The overall model was statistically significant, χ^2^(12) = 25.5, *p* = 0.013, with a McFadden’s R^2^ of 0.376, indicating a moderate effect size (Table 4).

Logistic regression identified female gender, higher barrier scores, and lower knowledge scores as predictors of decision change toward consent. However, the wide confidence intervals (e.g., AOR = 35.6, 95% CI 1.34–998.4 for female gender) indicate instability due to small subgroup sizes. These findings should therefore be interpreted as exploratory.

Barriers to organ donation were also a negative predictor of change: participants with higher baseline barrier scores were less likely to shift to consent (AOR = 0.05, *p* = 0.017). Economic well-being showed a similar inverse association (AOR = 0.16, *p* = 0.030), indicating that individuals with lower perceived economic well-being were more likely to change their decision.

Other variables, including school, age, language, place of residence, religion, and religiosity, were not significant predictors in the model. These findings underscore the importance of addressing informational and psychological barriers in educational interventions and highlight the possible role of gender and knowledge gain in facilitating attitudinal change toward organ donation within more ambivalent groups.

Moreover, among all the assessed barriers, the only significant difference between the DLCR-LC and DLCR-DLCR groups was found for the barrier related to not knowing how to express consent (t = 2.163, *p* = 0.036).

## 4. Discussion

This study provides preliminary evidence that a single, structured educational lecture may improve university students’ knowledge and positively influence their willingness to consent to postmortem organ donation. The findings have significant implications for public health strategies in Kazakhstan and similar sociocultural contexts, where low donor registration rates persist despite legal frameworks supporting organ donation.

Organ donation rates vary widely across the world and are typically measured as deceased donors per million population (pmp). Spain remains the global leader with 33–35 donors pmp, followed by France at 23.2 donors pmp, while the United States also ranks among the top performers under its opt-in system, though challenges in donor conversion persist [19,20,21]. At the regional level, the Americas report the highest number of transplants (62,153), followed by Europe (40,337) and the Western Pacific (29,014), whereas Africa recorded only 286 in 2022, almost exclusively from living donors [22]. Globally, most of the 108,818 transplanted organs originate from deceased donors, underscoring the contrast with Kazakhstan, where donation rates remain extremely low at 0.3 pmp in 2024 [23]. National registry data confirm this imbalance: between 2012 and 2023, Kazakhstan performed 92 heart transplants, 493 LVAD implantations, and 474 liver transplants (of which 411 (86.7%) relied on living donors) highlighting the country’s persistent dependence on living donation despite expanding surgical capacity [24,25].

The observed increase in knowledge scores and change in attitudes toward donation suggest that even brief interventions can break through ambivalence and uncertainty—two of the most persistent barriers to donor registration identified in the global literature. Previous studies in Western and Eastern Europe have shown that increased knowledge correlates with improved donor attitudes and intentions to register, especially among young people [26,27]. However, our study is among the first to demonstrate these effects in Central Asia, a region where sociocultural and religious dynamics pose unique challenges to public health messaging on organ donation.

One of the most encouraging findings was the significant shift in donation decisions among participants who had initially left the decision to family members (DLCR group). Within this subgroup, over 45% ultimately expressed willingness to donate after the lecture, suggesting the potential value of education for those who are undecided. Importantly, the logistic regression model revealed that knowledge gain was a strong, independent predictor of this attitudinal change (AOR = 3.03, *p* = 0.010), indicating an association between knowledge gain and attitudinal change in this context.

The data also highlight the psychological and procedural dimensions of decision-making. Among various perceived barriers, uncertainty about how to express consent was the single most important factor distinguishing participants who changed their minds from those who did not. This finding points to a specific and actionable gap: many people are not opposed to organ donation in principle but lack the information or confidence to formalize their decision. Addressing this procedural ambiguity—through clearer communication about consent registration pathways, standardized forms, and visible donation infrastructure—should be a core component of future public campaigns [28,29].

In addition, the study identified several moderators of knowledge gain and attitude change. Gender was a particularly influential factor: female participants were significantly more likely to shift toward consent (AOR = 35.59, *p* = 0.033). This is consistent with previous research suggesting that women are more likely to engage in health-promoting and altruistic behaviors, including donation [30,31]. From a programmatic perspective, this suggests that female students may serve as effective peer advocates in university-based interventions or public awareness campaigns.

Another important moderator was academic specialization. Non-medical students, despite having lower baseline knowledge scores, demonstrated greater relative improvement following the lecture. This highlights an untapped opportunity for donor education beyond medical or allied health schools. In particular, reaching students in the social sciences, humanities, and technical fields could yield significant returns, as these groups typically have limited exposure to bioethics or transplantation-related content in their curricula [32,33]. This finding also supports a more inclusive and horizontal approach to organ donation education that crosses disciplinary boundaries.

Interestingly, one of the more unexpected findings was that the highest rate of refusal at baseline was observed among participants from the AMU, the largest medical training institution in the country. This finding contradicts the assumption that medical students, by virtue of their education, would be more supportive of organ donation [27,34]. Several factors could explain this phenomenon. First, early medical students may be more aware of clinical complexities, uncertainties about transplant outcomes, or concerns about body integrity, which could increase hesitancy. Second, medical curricula in Kazakhstan may not provide systematic training in the ethics of organ donation, consent procedures, or transplantation policy—topics that are often left to postgraduate specialization. Third, exposure to critically ill or deceased patients during early clinical rotations may create emotional distance or reinforce perceptions of professional detachment rather than personal advocacy. This paradox highlights the need to critically evaluate how organ donation is framed and taught in medical education in Kazakhstan. Curricular reforms that humanize the donation process and emphasize its life-saving impact may help align professional training with public health goals [35,36,37].

The effects of language and institution further underscore the need for culturally and contextually sensitive outreach. Russian-speaking students and those from higher-ranked institutions such as NU had higher baseline literacy, while those from ENU showed the most dramatic gains. These patterns suggest that both baseline literacy and institutional emphasis on health ethics may shape receptivity to donor education [38,39,40]. Therefore, multi-institutional collaboration and bilingual delivery are essential to maximize reach and equity across diverse student populations.

Interestingly, lower perceived economic well-being was associated with a higher likelihood of shifting toward consent. While counterintuitive at first glance, this may reflect a greater sense of collective responsibility or greater resonance with solidarity-based health messages among economically disadvantaged groups [41,42]. Alternatively, individuals in this group may perceive fewer personal or family risks associated with donation, although this requires further qualitative investigation. Regardless of the interpretation, the association calls for nuanced messaging that addresses the ethical fairness of organ allocation and builds trust in the healthcare system [43].

The current study also sheds light on the complex relationship between religion, religiosity, and attitudes toward donation. Although religiosity was not a significant predictor in the final regression model, baseline group comparisons showed that individuals who identified as Muslim expressed more reluctance, consistent with findings in other predominantly Muslim countries [44,45]. However, the educational intervention still resulted in positive changes among religious participants, suggesting that faith-based concerns can be effectively addressed through respectful, evidence-based dialog. Collaboration with religious leaders and the inclusion of culturally tailored theological content in future sessions could enhance this effect [46,47].

Taken together, these findings underscore that effective fundraising campaigns must go beyond providing information [48]. They must address emotional, ethical, and procedural uncertainties while being sensitive to the demographic and cultural profiles of the target populations. The results of this study support several practical recommendations. First, incorporating donor education into university curricula—even as part of general ethics or civic education modules—can serve as a scalable model. Second, public messaging should clearly explain how to legally register consent, possibly through visual guides, digital tools, or “donor ID” systems. Third, special attention should be given to tailoring materials by gender, language, and educational level to ensure inclusive outreach.

Finally, the fact that only a small minority of participants (3%) changed their attitudes toward giving negatively suggests that such educational interventions have a minimal risk of backfire effects. This suggests that education-based approaches may represent a promising strategy to increase informed consent and normalize donation decisions, although further controlled studies are needed.

### 4.1. Policy Recommendations

The findings of this study highlight several actionable strategies to enhance public engagement with postmortem organ donation in Kazakhstan. First, the significant increase in knowledge and willingness to donate following a single educational lecture demonstrates the effectiveness of brief, structured informational interventions. Therefore, integrating donation-related content into university curricula, particularly in both medical and non-medical disciplines, could serve as a scalable approach to improve awareness and shape favorable attitudes among young adults.

Second, the identification of “not knowing how to express consent” as the only barrier significantly distinguishing those who changed their minds from those who did not underscores a critical informational gap. Public health campaigns should prioritize clear communication about the legal and practical steps required to register one’s donation decision. This could include promoting online consent registries, simplifying documentation processes, and ensuring visibility of consent mechanisms in healthcare settings.

Third, the observed disparities in knowledge gain and attitude change based on gender, language, and academic specialization suggest that communication strategies should be tailored to different subgroups. Materials in both Kazakh and Russian, using gender-sensitive messaging and non-technical language, may enhance reach and effectiveness. Since non-medical students demonstrated greater knowledge improvement, general public outreach, not only among health professionals, should be a key focus.

Finally, the negative association between economic well-being and willingness to donate highlights the need to address socioeconomic concerns and potential mistrust in the healthcare system. Ensuring transparency, ethical governance, and equity in organ allocation procedures may improve public confidence and increase consent rates across all socioeconomic groups.

These recommendations emphasize that improving organ donation rates requires more than biomedical infrastructure; it also depends on sustained, targeted, and culturally competent education and policy initiatives.

### 4.2. Study Limitations

Several limitations should be considered when interpreting the findings of this study. First, the study employed a quasi-experimental pre–post design without a control group, which limits the ability to attribute changes solely to the educational intervention. Thus, prior intervention studies have shown the methodological importance of including a control group. The absence of such a group in the present study is therefore a major limitation, and our findings should be interpreted with caution until replicated in controlled designs. While significant improvements in knowledge and willingness to donate were observed, these changes may also reflect short-term social desirability bias or heightened awareness immediately after the lecture rather than a sustained attitudinal shift. As such, the findings should be interpreted as preliminary evidence of association rather than definitive proof of effectiveness. Second, there was participant attrition between the pre- and post-test phases, with only 97 of the original 129 participants completing both assessments. This drop-out may have introduced selection bias if those who remained were more motivated or already more favorable toward organ donation. Third, the use of self-reported measures for knowledge, attitudes, and perceived barriers may be subject to social desirability bias, particularly following an educational session that may have influenced participants to provide more socially acceptable responses. Fourth, the study was limited to university students from three institutions located in a single city in Kazakhstan, and the sample was predominantly female (80%) with a high proportion of medical students. These characteristics substantially restrict the generalizability of the findings to the broader Kazakhstani population, and claims about national-level trends should therefore be interpreted with caution. Fifth, the survey did not capture whether participants had previously discussed organ donation with their families or were aware of their relatives’ views. Given that Kazakhstan operates under an opt-in system where families play a decisive role in the absence of a registered choice, this omission represents an additional limitation. Sixth, the study did not assess participants’ prior exposure to donation-related information through social media or whether they followed institutions active in this field. Given the importance of social networks among university students, this omission may have limited our ability to fully capture factors influencing attitudes. Seventh, the design of the educational program was relatively classical and primarily unidirectional. Although students were encouraged to ask questions, their interactions were not systematically evaluated, and active participation (e.g., student-created content or social media engagement) was not incorporated. Evidence from other contexts suggests that such participatory approaches may yield stronger improvements in knowledge, attitudes, and family communication, and this should be considered in future research. In addition, the logistic regression analyses should be interpreted with caution. The extremely wide confidence intervals reflect instability arising from small subgroup sizes and limit the precision of effect estimates. Moreover, multiple subgroup analyses were conducted without statistical correction, which increases the risk of type I error. These findings should therefore be considered exploratory and hypothesis-generating rather than confirmatory, and larger studies with pre-specified hypotheses are needed to validate them. Finally, the long-term impact of the intervention on actual behavior, such as registering as a donor or discussing donation wishes with family members, was not assessed. Future studies should include longitudinal follow-up to evaluate whether changes in knowledge and attitude translate into sustained behavioral change.

## 5. Conclusions

This study suggests that a single, structured educational lecture may enhance knowledge and positively influence attitudes toward postmortem organ donation among university students in Kazakhstan. The intervention appeared particularly impactful for ambivalent individuals, with nearly half of those in the “decision left to relatives” group expressing consent after the session. These findings highlight the potential of targeted, culturally appropriate education to reduce uncertainty and foster more informed donor decisions.

Key factors associated with positive attitudinal change included knowledge gain, female gender, and lower perceived informational and procedural barriers. The observation that uncertainty about how to express consent was the most salient barrier among those who changed their minds points to a critical but addressable gap in current public education strategies.

Moreover, the inclusion of non-medical students, who demonstrated the greatest knowledge improvement, underscores the value of extending donor education beyond medical cohorts. Tailoring outreach by gender, language, and academic background may further strengthen effectiveness. At the same time, our findings should be interpreted with caution, as the pre–post design without a control group and the potential influence of social desirability bias limit causal inference. While further randomized controlled trials and longitudinal studies are required to confirm causality and assess sustained behavioral outcomes, the present results provide preliminary evidence that brief, well-designed interventions can contribute to advancing informed consent and public engagement with organ donation.

## Figures and Tables

**Figure 1 healthcare-13-02483-f001:**
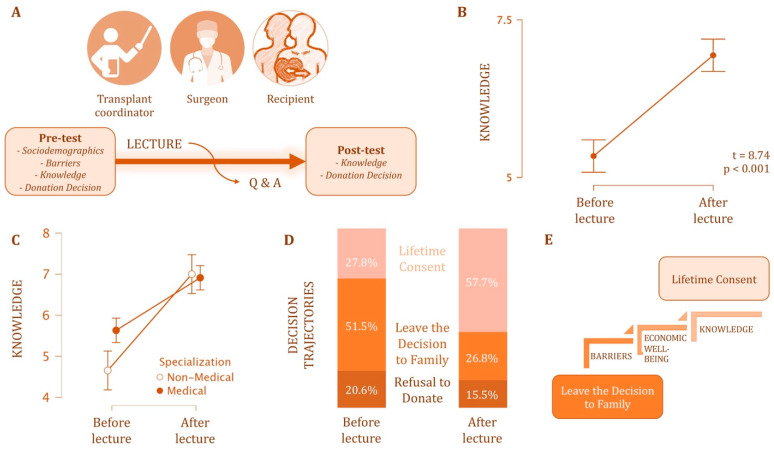
Impact of an educational lecture on students’ knowledge and attitudes toward postmortem organ donation (*n* = 129 pre-test; *n* = 97 post-test). (**A**) Overview of the study procedure: participants completed a pre-test assessing sociodemographic characteristics, perceived barriers, knowledge, and donation decisions. They then attended a standardized 60-min lecture delivered by three speakers (a transplant coordinator, a transplant surgeon, and a transplant recipient sharing personal experience), followed by a post-test. (**B**) Mean knowledge scores before and after the lecture (pre-test: M = 5.34 ± 1.67, post-test: M = 6.94 ± 1.33; t = 8.74, *p* < 0.001). (**C**) Knowledge improvement by academic specialization, with non-medical students demonstrating greater gains compared to medical students. (**D**) Distribution of donation decisions before and after the lecture: lifetime consent increased from 27.8% to 57.7%, while leaving the decision to family decreased from 51.5% to 26.8%. (**E**) Predictors of positive decision change from “decision left to relatives” to “lifetime consent,” including lower perceived barriers, higher economic well-being, and greater knowledge gain.

**Table 1 healthcare-13-02483-t001:** Study population of pre-test (*n* = 129) and post-test (*n* = 97).

Variable	Pre-Test *n* (%)/M ± SD	Post-Test *n* (%)/M ± SD	χ2/t, *p*
School			2.77, *p* = 0.251
NU ^1^	28 (21.7%)	15 (15.5%)	
ENU ^2^	43 (33.3%)	28 (28.9%)	
AMU ^3^	58 (45.0%)	54 (55.7%)	
Gender			0.001, *p* = 0.969
Male	25 (19.4%)	19 (19.6%)	
Female	104 (80.6%)	78 (80.4%)	
Age	18.8 ± 2.16	18.6 ± 1.69	0.981, *p* = 0.328
Specialization			1.06, *p* = 0.303
Non-medical	47 (36.4%)	29 (29.9%)	
Medical	82 (53.6%)	68 (70.1%)	
Language			0.838, *p* = 0.360
Kazakh	100 (77.5%)	80 (82.5%)	
Russian	29 (22.5%)	17 (17.5%)	
Residence before university enrolling			0.195, *p* = 0.659
Rural	39 (30.2%)	32 (33.0%)	
Urban	90 (69.8%)	65 (67.0%)	
Religion			0.658, *p* = 0.720
Islam	106 (82.2%)	81 (83.5%)	
Agnosticism	10 (7.8%)	5 (5.2%)	
Atheism	13 (10.1%)	11 (11.3%)	
Religiosity (1–5)	2.90 ± 1.12	2.86 ± 1.15	0.287, *p* = 0.774
Economic well-being (1–5)	3.57 ± 0.86	3.57 ± 0.88	0.010, *p* = 0.992
Knowledge on organ donation (0–8)	5.29 ± 1.82	5.34 ± 1.67	0.193, *p* = 0.847
Barriers of organ donation (1–5)	3.14 ± 0.66	3.19 ± 0.66	0.507, *p* = 0.612

^1^ NU—Nazarbayev University; ^2^ ENU—L.N. Gumilyov Eurasian National University; ^3^ AMU—Astana Medical University.

**Table 2 healthcare-13-02483-t002:** Group differences in sociodemographic characteristics, knowledge, and barriers according to pre-test donation attitudes (*n* = 129).

Variable	Lifetime Refusal (*n* = 25)	Decision Left to Close Relatives (*n* = 64)	Lifetime Consent (*n* = 40)	χ^2^/F, *p* Post Hoc
School				16.6, *p* = 0.002
NU ^1^	0	12 (42.9%)	16 (57.1%)	
ENU ^2^	10 (23.3%)	20 (46.5%)	13 (30.2%)	
AMU ^3^	15 (25.9%)	32 (55.2%)	11 (19.0%)	
Gender				4.68, *p* = 0.096
Male	2 (8.0%)	17 (68.0%)	6 (24.0%)	
Female	23 (22.1%)	47 (45.2%)	34 (32.7%)	
Age	18.4 ± 1.19	18.4 ± 1.08	19.9 ± 3.31	3.88, *p* = 0.027 ^4^
Specialization				0.719, *p* = 0.698
Non-medical	10 (21.3%)	21 (44.7%)	16 (34.0%)	
Medical	15 (18.3%)	43 (52.4%)	24 (29.3%)	
Language				8.70, *p* = 0.013
Kazakh	23 (23.0%)	52 (52.0%)	25 (25.0%)	
Russian	2 (6.9%)	12 (41.4%)	15 (51.7%)	
Residence before university enrolling				5.01, *p* = 0.082
Rural	12 (30.8%)	18 (46.2%)	9 (23.1%)	
Urban	13 (14.4%)	46 (51.1%)	31 (34.4%)	
Religion				7.45, *p* = 0.114
Islam	23 (21.7%)	55 (51.9%)	28 (26.4%)	
Agnosticism	0	5 (50.0%)	5 (50.5%)	
Atheism	2 (15.4%)	4 (30.8%)	7 (53.8%)	
Religiosity (1–5)	3.24 ± 1.05	2.94 ± 1.10	2.63 ± 1.15	2.46, *p* = 0.094
Economic well-being (1–5)	3.68 ± 0.80	3.52 ± 0.87	3.58 ± 0.87	0.36, *p* = 0.702
Knowledge on organ donation (0–8)	5.04 ± 1.54	5.00 ± 1.87	5.92 ± 1.79	3.58, *p* = 0.033 ^5^
Barriers of organ donation (1–5)	3.36 ± 0.68	3.28 ± 0.58	2.78 ± 0.61	10.0, *p* < 0.001 ^6^

^1^ NU—Nazarbayev University; ^2^ ENU—L.N. Gumilyov Eurasian National University; ^3^ AMU—Astana Medical University; ^4^ post hoc test—LC vs. LR/DLCR, *p* < 0.05; ^5^ post hoc test—LC vs. DLCR, *p* < 0.05, ^6^ Post hoc test: LC vs. LR/DLCR, *p* < 0.01.

**Table 3 healthcare-13-02483-t003:** Differences in knowledge about postmortem organ donation before and after the lecture across sociodemographic and educational subgroups.

Variable	Knowledge Before	F-Test, *p* Post Hoc	Knowledge After	Paired T-Test, *p*	Repeated ANOVA, F (Knowledge × Factor), *p*
School		14.81, *p* < 0.001 ^4^			4.13, *p* = 0.019
NU ^1^	6.73 ± 1.03		7.67 ± 0.62	2.07, *p* = 0.311	
ENU ^2^	4.61 ± 1.62		6.96 ± 1.17	7.15, *p* < 0.001	
AMU ^3^	5.33 ± 1.61		6.72 ± 1.46	5.85, *p* < 0.001	
Gender		0.38, *p* = 0.708			4.31, *p* = 0.041
Male	5.21 ± 2.10		6.05 ± 1.93	2.07, *p* = 0.169	
Female	5.37 ± 1.56		7.15 ± 1.05	8.89, *p* < 0.001	
Specialization		2.72, *p* = 0.008			7.61, *p* = 0.007
Non-medical	4.66 ± 1.61		7.00 ± 1.16	7.25, *p* < 0.001	
Medical	5.63 ± 1.62		6.91 ± 1.40	4.33, *p* < 0.001	
Language		3.03, *p* = 0.003			1.13, *p* = 0.290
Kazakh	5.11 ± 1.65		6.80 ± 1.38	8.39, *p* < 0.001	
Russian	6.41 ± 1.37		7.59 ± 0.79	2.69, *p* = 0.041	
Residence before university enrolling		0.630, *p* = 0.530			0.377, *p* = 0.541
Rural	5.19 ± 1.99		6.63 ± 1.74	4.50, *p* < 0.001	
Urban	5.42 ± 1.50		7.09 ± 1.06	7.48, *p* < 0.001	
Religion		5.71, *p* = 0.038			1.38, *p* = 0.257
Islam	5.14 ± 1.65		6.86 ± 1.38	8.67, *p* < 0.001	
Agnosticism	6.40 ± 1.14		7.60 ± 0.55	1.50, *p* = 0.668	
Atheism	6.36 ± 1.57		7.18 ± 1.17	1.51, *p* = 0.657	

^1^ NU—Nazarbayev University; ^2^ ENU—L.N. Gumilyov Eurasian National University; ^3^ AMU—Astana Medical University; ^4^ Post hoc test—NU vs. ENU/AMU, *p* < 0.001.

**Table 4 healthcare-13-02483-t004:** Logistic regression predicting decision change toward consent among participants Initially in the DLCR group (*n* = 49, χ^2^ = 25.5, *p* = 0.013, R^2^_McF_ = 0.376).

Variable	Beta	AOR	AOR 95% CI	*p*
School				
ENU ^2^–NU ^1^	0.271	1.311	0.034–50.02	0.884
AMU ^3^−NU ^1^	1.326	3.765	0.069–205.84	0.516
Gender				
Female−Male	3.600	35.59	1.341–998.41	0.033
Age	0.456	1.578	0.467–5.332	0.463
Language				
Kazakh−Russian	0.794	2.211	0.065–75.11	0.659
Residence before university enrolling				
Urban−Rural	0.335	1.397	0.181–10.76	0.748
Religion				
Agnosticism−Islam	2.160	8.671	0.099–756.52	0.343
Atheism−Islam	−0.357	0.700	0.002–199.53	0.902
Religiosity (1–5)	0.331	1.393	0.487–3.984	0.537
Economic well-being (1–5)	−1.813	0.163	0.032–0.837	0.030
Knowledge on organ donation (0–8)	−2.992	0.050	0.004–0.580	0.017
Barriers of organ donation (1–5)	1.109	3.033	1.307–7.036	0.010

^1^ NU—Nazarbayev University; ^2^ ENU—L.N. Gumilyov Eurasian National University; ^3^ AMU—Astana Medical University.

## Data Availability

The raw data supporting the conclusions of this article will be made available by the authors on request.

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
