# Peer review of "From Uncertainty to Consent: Educational Intervention Effects on Knowledge and Willingness to Donate Organs After Death"

_healthcare, 2025, doi:10.3390/healthcare13192483_

Round 1

Reviewer 1 Report

Comments and Suggestions for Authors

Thank you for the opportunity to review the manuscript. This quasi-experimental interventional study aimed to evaluate the impact of a single educational lecture on university students' knowledge of and attitudes toward deceased organ donation. The authors reported that the lecture was associated with an increase in both knowledge and willingness to donate organs after death. While this study is valuable and thought-provoking, several concerns warrant attention.

  • Considering the limitations of this study described in the limitations section, I would suggest that the authors town down the conclusion section. Only randomized controlled trials conclude that “the lecture increased both knowledge and willingness to donate organs after death.”

  • I would suggest that the authors add a table that includes all questions to evaluate the knowledge used in this study. This would be helpful for readers to understand the significance of this study.

  • In addition to the above comments, I suggest describing the languages used in the pre-test and post-test. Did the author prepare for the two types of tests in Kazakhs and Russians?

  • I would suggest that the authors clearly describe whether the odds ratio is unadjusted or adjusted throughout the manuscript. The term “odds ratio” may be ambiguous for readers to understand the results of this study.

  • To enhance the impact of this study, I would suggest that the authors cite two more valuable articles in the introduction section. The former was a review and the latter had a large sample size and used a validated questionnaire to evaluate the association between knowledge and willingness to donate organs.

#1 Article regarding worldwide barriers to organ donation (line 44)
Da Silva IR, et al. Worldwide barriers to organ donation. JAMA Neurol 2015;72(1):112-118.

#2 Article regarding association between knowledge and willingness to donate organs after death (Line 51)

Murakami M, et al. Knowledge does not correlate with behavior toward deceased organ donation: a cross-sectional study in Japan. Ann Transplant 2020;25:e918936.

  • I would suggest that the authors carefully recheck the reference list. For example, please describe the journal in the appropriate abbreviation for reference #29.

Author Response

General comment: Thank you for the opportunity to review the manuscript. This quasi-experimental interventional study aimed to evaluate the impact of a single educational lecture on university students' knowledge of and attitudes toward deceased organ donation. The authors reported that the lecture was associated with an increase in both knowledge and willingness to donate organs after death. While this study is valuable and thought-provoking, several concerns warrant attention.

Response: We sincerely thank the reviewer for their thoughtful evaluation of our manuscript and for recognizing the value of our quasi-experimental interventional study. We agree that while the study provides novel and contextually relevant insights into the role of educational interventions in shaping knowledge and willingness to donate organs, several aspects warrant careful consideration.

Comment 1: Considering the limitations of this study described in the limitations section, I would suggest that the authors town down the conclusion section. Only randomized controlled trials conclude that “the lecture increased both knowledge and willingness to donate organs after death.”

Response 1: We thank the reviewer for this important comment and fully agree that, given the quasi-experimental design, our conclusions should be expressed with greater caution. In line with this recommendation, we have revised the Conclusion section to tone down causal claims and emphasize associations rather than definitive effects. Specifically, we now use phrasing such as “may enhance,” “positively influence,” and “appeared particularly impactful” instead of “increased.” We also explicitly state that randomized and longitudinal studies are needed to establish causality and long-term impact.

Comment 2: I would suggest that the authors add a table that includes all questions to evaluate the knowledge used in this study. This would be helpful for readers to understand the significance of this study.

Response 2: We thank the reviewer for the suggestion. To improve clarity, we revised the Methods section to specify that both the knowledge and barrier items were taken directly from a previously published questionnaire [14], where the full wording is available. Since this scale has already been published in detail, we did not reproduce it in the present manuscript but provided the reference for readers to consult.

Comment 3: In addition to the above comments, I suggest describing the languages used in the pre-test and post-test. Did the author prepare for the two types of tests in Kazakhs and Russians?

Response 3: We thank the reviewer for this helpful observation. We agree that clarifying the languages of the pre- and post-test questionnaires is important for transparency. In the revised manuscript, we have added this information in two places:

  • Section 2.1 (Study Design and Participants): we now specify that the questionnaire was administered in both Kazakh and Russian, and participants could choose their preferred language.
  • Section 2.5 (Ethical Considerations): we note that providing the survey instruments in both languages ensured accessibility, comprehension, and respect for participants’ language preferences.

We hope these additions clarify that both the pre-test and post-test were available in Kazakh and Russian.

Comment 4: I would suggest that the authors clearly describe whether the odds ratio is unadjusted or adjusted throughout the manuscript. The term “odds ratio” may be ambiguous for readers to understand the results of this study.

Response 4: We thank the reviewer for highlighting this important point. In the revised manuscript, we have clarified that all odds ratios reported from logistic regression analyses are adjusted for the covariates included in the model. Specifically, we (a) updated the Statistical Analysis section to explicitly state that logistic regression yielded adjusted odds ratios (AORs), (b) modified table headings and footnotes accordingly, and (c) revised the Results text to use the term “Adjusted OR (AOR)” instead of the more ambiguous “OR.” We believe these changes improve the clarity and interpretability of our findings.

Comment 5: To enhance the impact of this study, I would suggest that the authors cite two more valuable articles in the introduction section. The former was a review and the latter had a large sample size and used a validated questionnaire to evaluate the association between knowledge and willingness to donate organs.

#1 Article regarding worldwide barriers to organ donation (line 44)

Da Silva IR, et al. Worldwide barriers to organ donation. JAMA Neurol 2015;72(1):112-118.

#2 Article regarding association between knowledge and willingness to donate organs after death (Line 51)

Murakami M, et al. Knowledge does not correlate with behavior toward deceased organ donation: a cross-sectional study in Japan. Ann Transplant 2020;25:e918936.

Response 5: We thank the reviewer for this constructive suggestion. In line with the recommendation, we have incorporated both references into the Introduction section to strengthen the background and contextual framing of our study.

Comment 6: I would suggest that the authors carefully recheck the reference list. For example, please describe the journal in the appropriate abbreviation for reference #29.

Response 6: We thank the reviewer for this observation. We have carefully re-checked the reference list, including reference #29. The journal title in that citation follows APA style, which requires the full journal name rather than an abbreviated form. As there is no recommended abbreviation for this journal title under APA guidelines, we have retained the full journal name to ensure compliance with the journal’s referencing requirements.

Professional English editing in progress.

Reviewer 2 Report

Comments and Suggestions for Authors This study addresses an important public health issue — willingness to donate organs after death — in a Central Asian context that is underexplored in the literature. The educational intervention is timely, well structured, and clearly demonstrates short-term improvements in knowledge and consent rates and I would like to congratulate the authors for this work. However, several methodological and reporting limitations reduce the strength of the conclusions.   MAJOR COMMENTS 1. Study Design and Bias
The pre–post design without a control group limits causal inference. Observed improvements could partly reflect social desirability bias after the lecture rather than a true attitudinal shift. Authors should acknowledge this limitation more explicitly and temper claims about effectiveness. 2. Attrition and Representativeness
Out of 129 participants, only 97 completed the post-test. Attrition is >25%, which raises risk of selection bias. Were those lost to follow-up systematically different (e.g., gender, specialty)? A comparison table of completers vs. non-completers would be useful. 3. Statistical Analysis and Reporting •Logistic regression shows extremely wide confidence intervals (e.g., OR = 35.6, CI 1.34–998.4). These suggest instability from small subgroup sizes. The authors should present this as exploratory and avoid strong causal language. •Multiple tests are reported across subgroups without correction. The risk of type I error should be acknowledged. 4. Over-interpretation of Results
In the Discussion (lines 347–451), the manuscript sometimes overstates the impact of a single lecture (e.g., “transformative potential,” “robustness of education-based approaches”). Given methodological constraints, conclusions should be framed more cautiously — as promising but preliminary evidence. 5. Generalizability
The sample is predominantly female (80%) and drawn from three universities in one city. This should be better emphasized as a limitation (lines 492–495). Claims about broader Kazakhstani society should be scaled back.   MINOR COMMENTS 1. Language and Grammar •Line 123: “Immediately afterward, they attended a 60-minute informational lecture…” – smoother phrasing: “They then attended a standardized 60-minute lecture…”
Overall, text is clear but occasionally verbose; tighter phrasing would improve readability. A professional proofreading pass is recommended. 2. Terminology Consistency
The categories “DLCR,” “LR,” and “LC” (lines 299–301) are confusing and introduced mid-results. Abbreviations should be explained clearly and consistently in Methods. 3. Figures and Tables •Figure 1 is referenced but not described in sufficient detail in text (lines 87–100). Captions should be self-contained, including sample sizes. •Table 1: Present both n and % in each cell for clarity. Currently percentages are provided inconsistently. 4. References and Formatting •References mix styles (e.g., line 553 lists full first names, while others use initials). Ensure adherence to journal format. •Some references are repeated in text unnecessarily (Bolatov et al., 2025a/b appear multiple times close together, lines 46–68). These could be consolidated. •Verify accuracy of all DOIs; several appear truncated in the current version (e.g., line 559–561). 5. Ethical Considerations
Although ethics approval is stated (lines 175–178), it would be useful to clarify whether the study was preregistered and whether participants received any debriefing on organ donor registration options after the lecture.

Author Response

General Comment: This study addresses an important public health issue – willingness to donate organs after death – In a Central Asian context that is underexplored in the literature. The educational intervention is timely, well structured, and clearly demonstrates short-term improvements in knowledge and consent rates and I would like to congratulate the authors for this work. However, several methodological and reporting limitations reduce the strength of the conclusions.  

Response: We sincerely thank the reviewer for recognizing the significance of our study and for their encouraging remarks regarding the novelty, structure, and relevance of the intervention. We fully acknowledge that several methodological and reporting limitations constrain the strength of our conclusions. We hope these revisions strengthen the manuscript while maintaining its contribution to the underexplored Central Asian context of organ donation.

MAJOR COMMENTS

Comment 1. Study Design and Bias. The pre–post design without a control group limits causal inference. Observed improvements could partly reflect social desirability bias after the lecture rather than a true attitudinal shift. Authors should acknowledge this limitation more explicitly and temper claims about effectiveness.

Response 1: We thank the reviewer for this important comment. We agree that the pre–post design without a control group limits causal inference, and that observed improvements could partly reflect social desirability bias rather than a true attitudinal shift. In response, we have (a) revised the Study Limitations section to explicitly acknowledge these issues, and (b) tempered the Conclusion to frame the findings as preliminary associations rather than definitive evidence of effectiveness. We believe these revisions provide a more balanced interpretation of our results.

Comment 2. Attrition and Representativeness. Out of 129 participants, only 97 completed the post-test. Attrition is >25%, which raises risk of selection bias. Were those lost to follow-up systematically different (e.g., gender, specialty)? A comparison table of completers vs. non-completers would be useful.

Response 2: We thank the reviewer for highlighting the important issue of attrition and representativeness. In the revised manuscript, we have addressed this in two ways. First, we clarified in the Results that Table 1 already presents comparisons between the baseline sample (N=129) and those who completed the post-test (N=97). Second, as recommended, we have added Supplementary Table 2, which directly compares baseline characteristics of completers and non-completers. As shown, the two groups were broadly comparable across most sociodemographic, religious, and psychosocial variables, although some differences by school, specialization, age, and language were observed. These additions provide readers with a clearer understanding of attrition patterns and potential sources of selection bias.

Comment 3. Statistical Analysis and Reporting:

- Logistic regression shows extremely wide confidence intervals (e.g., OR = 35.6, CI 1.34–998.4). These suggest instability from small subgroup sizes. The authors should present this as exploratory and avoid strong causal language.

- Multiple tests are reported across subgroups without correction. The risk of type I error should be acknowledged.

Response 3: We thank the reviewer for this important comment. In the revised manuscript, we have tempered the presentation of our regression results. Specifically, we now (a) note in the Results that wide confidence intervals suggest instability due to small subgroup sizes, and (b) add to the Study Limitations that these analyses are exploratory, with the risk of type I error due to multiple testing. We emphasize that these findings are hypothesis-generating and require validation in larger, more robust studies.

Comment 4. Over-interpretation of Results. In the Discussion (lines 347–451), the manuscript sometimes overstates the impact of a single lecture (e.g., “transformative potential,” “robustness of education-based approaches”). Given methodological constraints, conclusions should be framed more cautiously — as promising but preliminary evidence.

Response 4: We thank the reviewer for pointing out the risk of over-interpretation. In response, we carefully revised the Discussion to frame the findings more cautiously. Specifically, we replaced stronger terms such as “compelling evidence,” “transformative potential,” “direct impact,” and “robustness” with softer language such as “preliminary evidence,” “potential value,” “association,” and “promising approach.” We also emphasized that the results should be viewed as exploratory given the study’s methodological constraints.

Comment 5: Generalizability. The sample is predominantly female (80%) and drawn from three universities in one city. This should be better emphasized as a limitation (lines 492–495). Claims about broader Kazakhstani society should be scaled back.  

Response 5: We thank the reviewer for this important observation. In the revised manuscript, we have strengthened the description of the study’s limited generalizability. Specifically, in the Study Limitations section we now emphasize that the sample was drawn from only three universities in a single city, was predominantly female (80%), and included a high proportion of medical students. We note that these characteristics substantially restrict the applicability of our findings to the broader Kazakhstani population and therefore caution against making national-level claims. We believe this revision provides a clearer and more balanced interpretation of the study’s scope and limitations.

MINOR COMMENTS

Comment 1: Language and Grammar •Line 123: “Immediately afterward, they attended a 60-minute informational lecture…” – smoother phrasing: “They then attended a standardized 60-minute lecture…” Overall, text is clear but occasionally verbose; tighter phrasing would improve readability. A professional proofreading pass is recommended.

Response 1: We thank the reviewer for this helpful suggestion. In the revised manuscript, we have adopted the recommended smoother. We also confirm that the manuscript has undergone a professional English editing service to improve readability and address issues of verbosity.

Comment 2: Terminology Consistency. The categories “DLCR,” “LR,” and “LC” (lines 299–301) are confusing and introduced mid-results. Abbreviations should be explained clearly and consistently in Methods.

Response 2: We thank the reviewer for this helpful suggestion. In the revised manuscript, we clarified terminology by introducing the abbreviations LC (lifetime consent), LR (lifetime refusal), and DLCR (decision left to close relatives) in the Methods section (2.4.4 Donation Decision). These changes ensure consistent and transparent use of terminology throughout the manuscript.

Comment 3. Figures and Tables.

- Figure 1 is referenced but not described in sufficient detail in text (lines 87–100). Captions should be self-contained, including sample sizes.

- Table 1: Present both n and % in each cell for clarity. Currently percentages are provided inconsistently.

Response 3: We thank the reviewer for this helpful suggestion. In the revised manuscript, we expanded the caption for Figure 1 to be self-contained, including sample sizes and clear explanations of all panels (A-E). We also added a brief descriptive sentence in the main text (Methods section) to highlight what Figure 1 illustrates. In the revised manuscript, we have reformatted Table 1 to present both the absolute number of participants (n) and the corresponding percentage (%) for each category. This ensures consistency across all variables and improves clarity for readers.

Comment 4. References and Formatting. References mix styles (e.g., line 553 lists full first names, while others use initials). Ensure adherence to journal format.

- Some references are repeated in text unnecessarily (Bolatov et al., 2025a/b appear multiple times close together, lines 46–68). These could be consolidated.

- Verify accuracy of all DOIs; several appear truncated in the current version (e.g., line 559–561).

Response 4: We thank the reviewer for pointing out the issues with references and formatting. In the revised manuscript, we have standardized all in-text citations to the numerical format (e.g., [1]) and ensured consistency across the reference list (using initials rather than full first names, in accordance with journal requirements). We also consolidated repeated references (e.g., Bolatov et al., 2025a/b) to avoid redundancy and improve readability. The truncated appearance of some DOIs was due to the submission template.

Comment 5. Ethical Considerations. Although ethics approval is stated (lines 175–178), it would be useful to clarify whether the study was preregistered and whether participants received any debriefing on organ donor registration options after the lecture.

Response 5: We thank the reviewer for this helpful suggestion. In the revised manuscript, we clarified the ethical procedures. Specifically, we noted in the Ethical Considerations section that the study was not preregistered, as it was exploratory in design, and that participants received a debriefing after the post-test with information on organ donor registration options. We believe this additional detail improves the transparency of our methodology.

Reviewer 3 Report

Comments and Suggestions for Authors

The manuscript is well-written. The aim of the study is novel, precise, and essential for regional transplantation programs. The design of the study is appropriate, and the statistical tools were chosen correctly. The results are clearly presented, and they support the conclusions. 

I have a few minor remarks to consider by the Authors, as follows:

  1. line 36 - p = .036) should be p = 0.036
  2. Please carefully evaluate the form of citations in the text. 
  3. lines 90-100 - is it part of the main text or the subtitle of the figure?
  4. More information about the informational lecture should be provided, e.g. as supplementary information. 
  5. AMU Astana Medical University (AMU)
  6. The discussion is comprehensive, but highlighting some information about statistics (worldwide vs Kazakhstan will be valuable.

Author Response

General comment: The manuscript is well-written. The aim of the study is novel, precise, and essential for regional transplantation programs. The design of the study is appropriate, and the statistical tools were chosen correctly. The results are clearly presented, and they support the conclusions.  I have a few minor remarks to consider by the Authors, as follows:

Response: We sincerely thank the reviewer for their positive evaluation of our work. We are encouraged by the recognition that the study aim is novel and relevant, the design appropriate, and the statistical tools correctly applied. We also appreciate the acknowledgment that the results are clearly presented and support the conclusions. We have carefully addressed the reviewer’s minor remarks point by point in the revised manuscript, and we believe these changes have further improved the clarity and rigor of our paper.

Comment 1: line 36 - p = .036) should be p = 0.036

Response 1: We thank the reviewer for noting this formatting issue. We have corrected the value at line 36 to “p = 0.036” and carefully re-checked the entire manuscript to ensure that all p-values are consistently reported in the correct format.

Comment 2: Please carefully evaluate the form of citations in the text.

Response 2: Thank you for the note. We carefully audited and standardized all in-text citations to match the journal’s guidelines.

Comment 3: lines 90-100 - is it part of the main text or the subtitle of the figure?

Response 3: We thank the reviewer for pointing out this formatting ambiguity. The text in lines 90-100 is the figure legend for Figure 1, not part of the main text. We have adjusted the formatting in the revised manuscript to make this clearer and consistent with the journal’s style.

Comment 4: More information about the informational lecture should be provided, e.g. as supplementary information.

Response 4: We thank the reviewer for this helpful suggestion. In response, we prepared a supplementary table (Supplementary Table 1) summarizing the structure and content of the educational lecture. The table outlines the major sections, key elements, and the speakers involved. We believe this addition enhances transparency and reproducibility of the intervention.

Comment 5: AMU – Astana Medical University (AMU)

Response 5: We thank the reviewer for noticing this redundancy. In the revised manuscript, we corrected the abbreviation in the table footnotes to read “AMU – Astana Medical University” (without repeating the abbreviation in parentheses).

Comment 6: The discussion is comprehensive, but highlighting some information about statistics (worldwide vs Kazakhstan will be valuable).

Response 6: We thank the reviewer for this valuable suggestion. In the revised Discussion section, we added a new paragraph highlighting worldwide versus Kazakhstan statistics on organ donation and transplantation. These additions provide clearer international and national context for interpreting the relevance of our educational intervention.

Reviewer 4 Report

Comments and Suggestions for Authors

I would like to thank the editor of this prestigious journal for the opportunity to evaluate this study and I would also like to congratulate the researchers for their efforts and the results obtained. In the field of organ donation and transplantation, it has been shown that the mere fact of receiving information already has a positive influence on the attitude towards this process in which the lives of people with serious illnesses can be saved and whose pathology can only be solved with an organ.

I would like to point out some issues that I believe can improve this study.

Firstly, numerous studies have been carried out on the impact of educational interventions, both on secondary school and university students, and they are not indicated in the theoretical framework, especially by Spanish research teams. In addition, these studies include a control group, which is absolutely necessary in order to know the influence of the proposal. This study, as its limitations indicate, does not have a control group, which is a very important deficiency.

In the theoretical framework it is imperative to develop the legal concept, i.e. whether the country where the educational proposal is made has presumed consent or informed consent (opt in - opt out). This is imperative because if the country is governed by opt-in, such as the US, the focus should be on the individual's attitude towards him/herself, including inviting the individual to register as a donor. On the other hand, when the system is opt-out, as in Spain, it is necessary to indicate in the educational intervention that it is necessary to know the opinion of our relatives and that our relatives know ours. This study, in its design, does not address this question in its forms on whether participants have discussed the issue with their family or whether they know their opinion.

The theoretical framework also does not address whether participants have received information about donation on social media or whether they are followers of any institution on the topic. If we are talking about information and in university students, who are the biggest consumers of social networks, it is necessary to address the issue in the theoretical framework and indicate it in the limitations of the study as well. A recent study of university health science students indicates the great importance of social media in the process of organ donation and the need to reach the general public with informative messages.

A section on the design of the educational intervention is missing in the material and methods section. A study derived from an international congress that brought together the most advanced countries in the process of organ donation and transplantation, made a series of indications to be taken into account when carrying out this type of intervention.

https://doi.org/10.1111/j.1399-3046.2012.01785.x

They also say that students belong to medical and non-medical programmes, what percentage of each? The information and attitude of a medical or nursing student is very different from that of other specialties, this is a bias for the study, you cannot indicate only generalist university students.

The design of the educational programme is very classical. It is unidirectional, the student does not interact and his or her questions are not evaluated or are residual in the study. In Spain, there have been educational programmes where students create content on donation and are evaluated, as well as sharing it on social networks and analysing the impact. When the student is an active part of the educational programme and not merely a recipient of information, the results in terms of attitude, knowledge and socio-familial interaction are much higher. This should be noted in the limitations of the study.

As for the forms administered, are they validated by a group of experts? It would be interesting to answer this and even to show the questions asked as an annex.

There is a mistake in the numbering of the theoretical framework, point 2.5 is repeated, Ethical considerations should be 2.6.

I congratulate the researchers for the excellent research they have carried out and I am grateful for the opportunity to have been able to evaluate this study.

Author Response

General comment: I would like to thank the editor of this prestigious journal for the opportunity to evaluate this study and I would also like to congratulate the researchers for their efforts and the results obtained. In the field of organ donation and transplantation, it has been shown that the mere fact of receiving information already has a positive influence on the attitude towards this process in which the lives of people with serious illnesses can be saved and whose pathology can only be solved with an organ. I would like to point out some issues that I believe can improve this study.

Response: We sincerely thank the reviewer for their thoughtful and encouraging remarks, as well as for recognizing the importance of our work. We fully agree that providing accurate information can have a meaningful influence on attitudes toward organ donation and transplantation, a field where awareness directly translates into lifesaving opportunities. We greatly appreciate the reviewer’s acknowledgement of our efforts and the study’s results, and we have carefully considered the issues raised in the detailed comments. We believe that the revisions made in response to these suggestions have strengthened the manuscript and improved its clarity, contextualization, and impact.

Comment 1: Firstly, numerous studies have been carried out on the impact of educational interventions, both on secondary school and university students, and they are not indicated in the theoretical framework, especially by Spanish research teams. In addition, these studies include a control group, which is absolutely necessary in order to know the influence of the proposal. This study, as its limitations indicate, does not have a control group, which is a very important deficiency.

Response 1: We thank the reviewer for this valuable observation. In the revised Introduction, we have strengthened the theoretical framework by adding references to Spanish studies that evaluated the impact of educational interventions among both secondary school and university students. These include randomized controlled trials, innovative school-based programs, and nurse-led workshops, all of which demonstrated significant improvements in knowledge, correction of misconceptions, and more favorable attitudes toward donation (Bas-Sarmiento et al., 2023; Febrero et al., 2021; Nieto-Galván et al., 2022). We fully agree with the reviewer that the absence of a control group is a major limitation of our study. This point is now emphasized more explicitly in the Study Limitations section, where we note that prior Spanish research highlights the methodological importance of including a control group, and that our findings should be interpreted as preliminary evidence until validated in controlled designs.

Comment 2: In the theoretical framework it is imperative to develop the legal concept, i.e. whether the country where the educational proposal is made has presumed consent or informed consent (opt in - opt out). This is imperative because if the country is governed by opt-in, such as the US, the focus should be on the individual's attitude towards him/herself, including inviting the individual to register as a donor. On the other hand, when the system is opt-out, as in Spain, it is necessary to indicate in the educational intervention that it is necessary to know the opinion of our relatives and that our relatives know ours. This study, in its design, does not address this question in its forms on whether participants have discussed the issue with their family or whether they know their opinion.

Response 2: We thank the reviewer for this important observation. In the revised Introduction, we have now developed the legal context of organ donation in Kazakhstan, clarifying that the country operates under an opt-in system, where individuals must actively register their decision through healthcare facilities or the national e-government portal. In the absence of such a decision, families are asked to decide on behalf of the deceased, making communication of one’s wishes to relatives particularly important. We also added recent national statistics to highlight the very low rate of registered consents. We acknowledge that our study design did not include specific questions on whether participants had previously discussed donation with their families or were aware of their relatives’ views. This represents an additional limitation, which we have now noted in the Study Limitations section.

Comment 3: The theoretical framework also does not address whether participants have received information about donation on social media or whether they are followers of any institution on the topic. If we are talking about information and in university students, who are the biggest consumers of social networks, it is necessary to address the issue in the theoretical framework and indicate it in the limitations of the study as well. A recent study of university health science students indicates the great importance of social media in the process of organ donation and the need to reach the general public with informative messages.

Response 3: We thank the reviewer for raising this important point. In the revised Introduction, we now acknowledge the growing influence of social media as a source of health information and its specific role in shaping university students’ attitudes toward organ donation. We also added a sentence to the Study Limitations section noting that our survey did not assess participants’ exposure to donation-related information on social networks or institutional accounts, which represents an additional limitation of the study.

Comment 4: A section on the design of the educational intervention is missing in the material and methods section. A study derived from an international congress that brought together the most advanced countries in the process of organ donation and transplantation, made a series of indications to be taken into account when carrying out this type of intervention. https://doi.org/10.1111/j.1399-3046.2012.01785.x

Response 4: We thank the reviewer for this important observation. In the revised manuscript, we added a detailed description of the design and delivery of the educational intervention in Section 2.3 (Procedure). This new subsection outlines the structure, language, content, and speaker composition of the standardized 60-minute lecture, including contributions from a transplant coordinator, transplant surgeon, and a transplant recipient. We also indicated that the intervention was developed in line with international recommendations from the First Global Forum on Education on Organ Donation and Transplantation (Cantarovich et al., 2012), which emphasize short, structured, and culturally sensitive activities involving both healthcare professionals and patient voices.

Comment 5: They also say that students belong to medical and non-medical programmes, what percentage of each? The information and attitude of a medical or nursing student is very different from that of other specialties, this is a bias for the study, you cannot indicate only generalist university students.

Response 5: We thank the reviewer for this valuable comment. In the revised manuscript, we clarified in Section 2.1 (Study Design and Participants) that all medical students were enrolled in the General Medicine/MBBS or MD programs. The distribution of medical and non-medical students is already presented in Table 1.

Comment 6: The design of the educational programme is very classical. It is unidirectional, the student does not interact and his or her questions are not evaluated or are residual in the study. In Spain, there have been educational programmes where students create content on donation and are evaluated, as well as sharing it on social networks and analysing the impact. When the student is an active part of the educational programme and not merely a recipient of information, the results in terms of attitude, knowledge and socio-familial interaction are much higher. This should be noted in the limitations of the study.

Response 6: We thank the reviewer for this insightful comment. In the revised Limitations section, we now explicitly acknowledge that the educational program followed a relatively classical, unidirectional format. While students were able to ask questions, their interactions were not systematically evaluated, and the intervention did not include participatory components such as student-generated content or social media engagement. We also note that evidence from other contexts suggests more interactive approaches may yield stronger outcomes, and this should be considered in future studies.

Comment 7: As for the forms administered, are they validated by a group of experts? It would be interesting to answer this and even to show the questions asked as an annex.

Response 7: We thank the reviewer for this important comment. The survey instruments were assessed by a panel of experts in transplant coordination, transplantation surgery, public health, medical education, and social psychology to ensure content validity and contextual relevance. All data regarding the survey instruments, including links to the scales from previously published papers, are presented in the Methods section. Given this, we did not add the full list of questions as an annex, but we refer readers to the cited original validated instruments for the exact wording of items.

Comment 8: There is a mistake in the numbering of the theoretical framework, point 2.5 is repeated, Ethical considerations should be 2.6.

Response 8: We thank the reviewer for carefully noting this formatting error.

Comment 9: I congratulate the researchers for the excellent research they have carried out and I am grateful for the opportunity to have been able to evaluate this study.

Response 9: We sincerely thank the reviewer for their kind words and recognition of our work. We greatly appreciate the time and thoughtful feedback provided during the review process, which has helped us to strengthen and improve the manuscript.

Professional English editing in progress.

Round 2

Reviewer 1 Report

Comments and Suggestions for Authors

Thank you for revising the manuscript. It has significantly improved from the original submission, and the authors deserve commendations for their continued efforts. I have no further comments.

Reviewer 2 Report

Comments and Suggestions for Authors

All necessary changes have been incorporated in the revisions, and all review points have been satisfactorily addressed. I thank the authors for their thorough work and consider the manuscript suitable for publication at the editor’s discretion.

Reviewer 4 Report

Comments and Suggestions for Authors

I consider that the authors have corrected the indications correctly, although the improvements cannot correct structural errors of the study, at least they are visible in limitations and this is useful for future references by other researchers.